

# Calcification detection on upper extremity arteries: a comparison of ultrasonic and X-ray methods

Yanli Yang[1,*], Na Lin[1,*], Yuankai Xu[2], Zheli Niu[1], Fulei Meng[1], Kaidi Zhang[1], Yuhuan Wang[3], Lin Ruan[1] and Lihong Zhang[1]

[1] Department of Nephrology, The First Hospital of Hebei Medical University, Shijiazhuang, China
[2] Department of Nephrology, Zhejiang Hospital, Hangzhou, China
[3] Department of Nephrology, The First Hospital of Shijiazhuang City, Shijiazhuang, China
[*] These authors contributed equally to this work.

## ABSTRACT

**Background.** Vascular calcification (VC) has been observed in patients with hemodialysis, whereas few studies have investigated calcification in the upper extremity vasculature. Both ultrasound and X-ray are used to investigate the calcification of arteries in patients. However, there is a lack of data on the consistency between these two methods. The aim of this study was to investigate the occurrence of VC in the radial and ulnar arteries of hemodialysis patients and investigate the detection consistency in VC between ultrasound and X-ray.

**Methods.** Ultrasound and X-ray examinations were performed in the radial and ulnar arteries of both the left and right upper extremities of 40 patients on hemodialysis. The calcification status of arteries was evaluated by the calcification index from ultrasound and X-ray respectively. Clinical variables of patients were collected from all the involved patients.

**Results.** Of the 40 patients, VC was detected in 31 patients by ultrasound, while X-ray detected VC in 22 patients. Compared to ultrasound assessment, X-ray assessment was 73.21% sensitive but only 66.35% specific with a positive predictive value of 53.95% for detecting calcifications in the radial or ulnar artery. The level of agreement between ultrasound and X-ray results was fair. In addition, our data showed that more ulnar arteries had VCs than the corresponding radial arteries.

**Conclusion.** Ultrasound is more sensitive in detecting the presence of calcified atherosclerotic lesions. Ultrasound and X-ray exhibited fair consistency. Ultrasound screening for upper extremity radial and ulnar arteries in hemodialysis patients may deserve attention to explore its clinical significance.

## INTRODUCTION

Vascular calcification (VC) is a very common phenomenon in hemodialysis patients and can result in elasticity loss in an artery and an increase in pulse wave velocity (*Toussaint & Kerr, 2007*). The classical pathology of VC in hemodialysis patients involves the intimal

Corresponding authors
Lin Ruan, ruanlintt@163.com
Lihong Zhang, cpx1998@sina.com

layer or the medial layer of the vessel wall. Intimal calcification is generally accompanied by an inflammatory response, which commonly can result in occlusive plaque formation, representing an advanced stage of atherosclerosis. Mineral deposition can be observed in the calcification of the medial layer of muscle-type conduit arteries as a pathological change independent of atherosclerosis. Moreover, both of these pathological changes can exist in the same vasculature simultaneously (*Falaknazi et al., 2012*; *Ku et al., 2006*; *McCullough et al., 2008*).

VC in hemodialysis patients is usually correlated with multiple clinical events, such as myocardial ischemia (*Chen et al., 2017*), stroke (*Power et al., 2011*) and vascular event-related mortality (*Gungor et al., 2018*). Therefore, identifying VC in hemodialysis patients as early as possible may help patients obtain suitable treatments and improve their survival and quality of life. Currently, multiple imaging techniques are available to screen for the presence of VC: plain X-rays, two-dimensional ultrasound, and computer tomography (CT). CT and X-ray will require the patient to undergo radiation exposure. CT can identify VC with high sensitivity, but it is an expensive examination. X-rays can identify calcifications in the aorta and peripheral arteries with high cost-effectiveness and are widely used in the clinic. Under X-ray, VC appears as an X-ray-negative plaque lining the vessel walls, either in an irregular pattern or in a linear pattern. Ultrasound is nonradioactive, cost-effective, and has minimal discomfort, and it might be an ideal option to screen VC in patients. Calcification of the artery can be shown as a hyperechogenic spot with a posterior shadow (*Jashari et al., 2015*). Ultrasound examination is considered the first choice for screening carotid artery calcification (*Johri et al., 2020*). Moreover, several studies have shown that ultrasound can provide good sensitivity and specificity in the diagnosis of lower extremity arterial disease (*Franz et al., 2013*; *Verim & Tasci, 2013*).

However, there is still a lack of data comparing the feasibility and accuracy of ultrasound and X-ray in VC identification in hemodialysis patients. Currently, the 2017 Kidney Disease Improving Global Outcomes guidelines weakly recommend performing routine screening for vascular calcification by using X-rays or CT in all patients with end-stage chronic kidney (ESCKD) at the time of dialysis initiation due to a lack of research data support (*Ketteler et al., 2018*). Thus, in the present study, we examined the upper extremity arteries of hemodialysis patients to screen and identify VC using both ultrasound and X-ray, compared the examination results between X-ray and ultrasound, and investigated the consistency and difference of these two examination methods.

## MATERIALS & METHODS

This was a single-center perspective study conducted from January 2021 to May 2021. The study included patients with ESCKD who had an arteriovenous fistula (AVF) for hemodialysis access. All enrolled patients had been maintained on stable hemodialysis (three times per week in 4 h sessions) for more than 3 months in our center (Fig. 1). Signed written consent forms agreeing to participate in the study were obtained from all the involved patients. The protocols of this study were approved by the First Hospital of Hebei Medical University institutional ethics committee (IRB NO.20210372, Chinese Clinical

Trial Registry No. ChiCTR1900021975). All methods were performed in accordance with the relevant guidelines and regulations.

## Sample size

The sample size was calculated by estimating 10 patients, considering $\alpha = 0.050$ (type I error) and $\beta = 0.200$ (type II error), which showed that a minimum of 38 patients was required to estimate the consistency between the two imaging methods in the study. Moreover, we reviewed previously published literature to determine the sample size (*Khosropanah et al., 2009*; *Yadav et al., 2020*; *Yu et al., 2021*). Based on previously published data and calculation results, 40 patients were recruited to estimate the consistency between the two imaging methods in the study.

### Inclusion and exclusion criteria

The inclusion criteria of this study included (1) ESCKD with stable hemodialysis: (three times per week in 4 h sessions) for more than 3 months; (2) age >18 years; (3) forearm AVF; and (4) signed written consent forms agreeing to participate in the study.

The exclusion criteria of the study included (1) systemic infection; (2) serious cardiac insufficiency; (3) failure to provide written consent; (4) involuntary upper limb extremity motions; (5) peripheral upper limb artery disease, including digital necrosis, change in skin color or temperature, infection/pain/reduced movement of the upper limb; (6) failure to complete either imaging examination and (7) suboptimal imaging data: ultrasound or X-ray.

In total, 59 patients were recruited in this study. However, four patients failed to provide a consent form and were excluded; three patients were excluded due to peripheral upper limb artery disease; one patient was excluded due to heart failure; and 11 patients were excluded due to failure to complete imaging examination or had suboptimal imaging data. Ultimately, 40 patients participated in this study.

## X-ray examination

On the posterior-anterior and lateral positions, plain X-rays of the upper extremities were obtained using a radiograph machine (MULTI X fusion Max, Siemens, Munich, Germany), which was then used to evaluate the VC in the radial and ulnar arteries. Any lesion lining the vessel walls with a high-density shadow, either in the intima with a thick, patchy or irregular pattern or in the media with a tubelike, regular, continuous or thin pattern, was considered vascular calcification (*Orr et al., 1978*).

To identify VC, all radiographic films were analyzed by one radiologist and one nephrologist. In different readings, the decision was made on the basis of consensus.

X-ray images were used to quantitatively assess severity by application of a score system in each patient. Every artery (radial and ulnar arteries in each arm) was divided equally into three segments, and then the VC was quantified based on each segment presentation as follows: 0, absence of calcification; 1, single punctate calcification; 2, multiple punctate lesions of calcification, or ≤3 lesions with irregular pattern & linear pattern (<6 cm); and 3, single lesion with linear calcification lesion ≥6 cm, or >3 lesions with irregular pattern & linear pattern (<6 cm). The final sum of points for each vascular segment generated

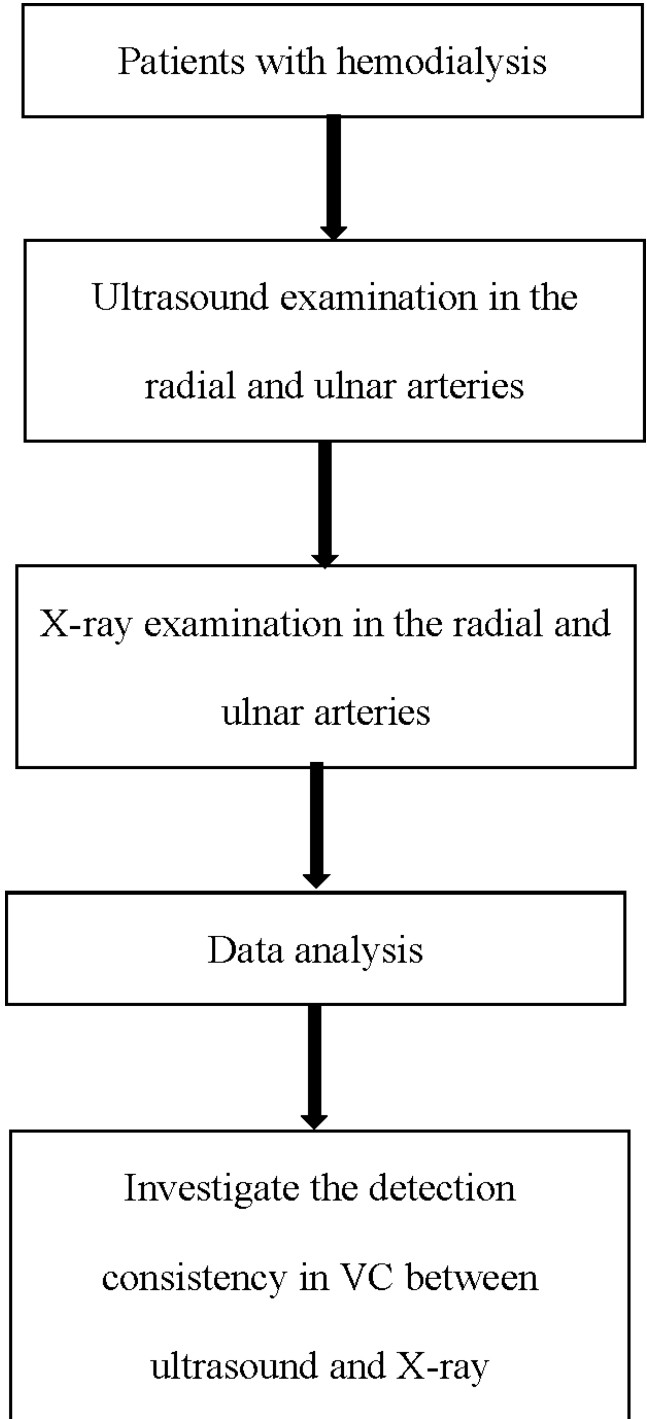

**Figure 1** **Flow chart of this study.**

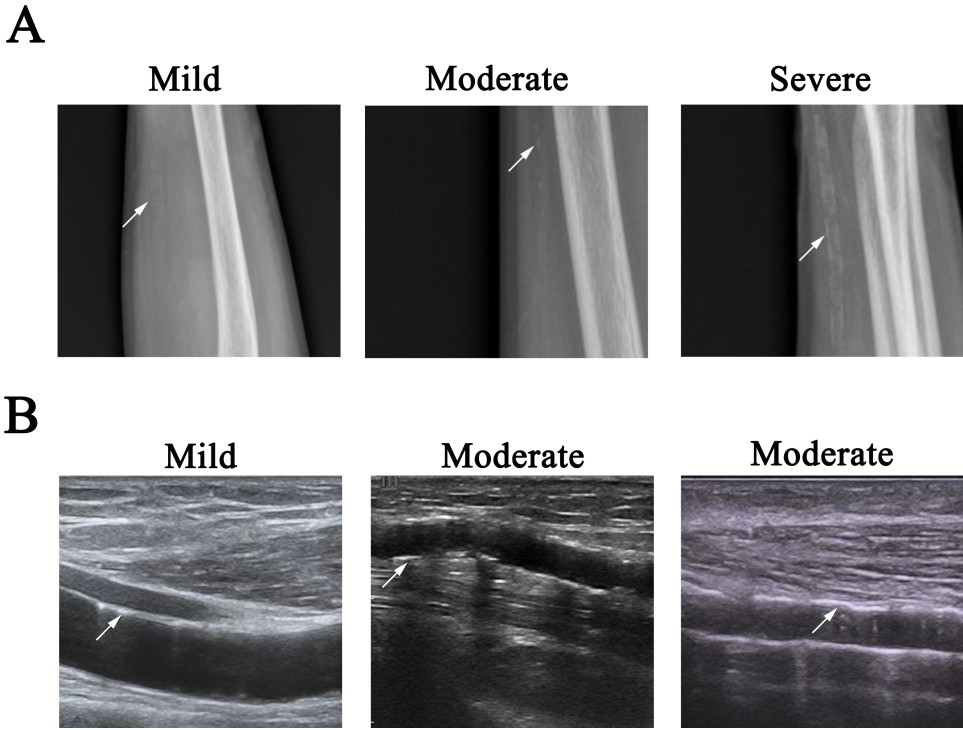

**Figure 2  Representative images of VC.** (A) Radio films showed mild, moderate, and severe arterial calcification, as shown by arrow. Left, single punctate calcification in ulnar artery; middle, lesion with linear pattern (1.7 cm in length) in ulnar artery; right, multiple lesions with irregular pattern & linear pattern in ulnar artery (the lesion label with arrow was about 3.9 cm in length). (B) Longitudinal B mode ultrasound arterial scan showed mild and moderate arterial calcification, as shown by arrow. Left: single lesion with 0.82 cm in length in the media of radial artery; middle: multiple linear lesions of calcification in the intima of ulnar artery (the lesion label with arrow was about 0.74 cm in length); right, multiple lesions in the media of radial artery (the lesion label with arrow was about 2.8 cm in length).

the calcification score of the artery, ranging from 0 to 9 points for each artery (Fig. 2A) (*Georgiadis et al., 2015*).

We use the following criteria to differentiate the VC severity of an artery: mild, any segment of the vascular had the score of 1, and the total score of an artery was $\leq 2$; moderate, any segment of the vascular had the score of 2, or the total score of an artery was 3 or 4 ($3 \leq$ total score $\leq 4$); severe, any segment of the vascular had the score of 3, or the total score of an artery was more than 4, then the entire vascular was considered severely diseased.

## Ultrasound examination

A Doppler diagnostic ultrasound system (Resona7s, Shenzhen, China) with linear array transducers (L20-5u 13.5 MHz) was used to assess the radial and ulnar arteries on both the left and right upper extremities. In brief, participants were placed in the sitting position, and B-mode ultrasound was used to observe the anatomy and determine VC lesion presence. The arteries of both sides were scanned in the cross-sectional plane to obtain an overall view of the blood vessels and detect the calcifications. Then, the arteries were scanned on

transverse and longitudinal sections from the origins of the radial and ulnar arteries to their distal parts, and the degree and type of calcifications were assessed. A hyperechogenic localized echo structure(inhomogeneous, spotty, irregular, adluminal) in the intima with posterior shadowing lining the vessel walls or a hyperechogenic localized echo structure with uniform, smooth, linear, and nonstenotic patten in the medial was considered vascular calcification (*Liu et al., 2012*).

The VC was quantified by application of the scoring system. Every artery (radial and ulnar arteries in each arm) was divided equally into three segments, and then the VC was quantified based on each segment presentation as follows: 0, absence of calcification; 1, single punctate calcification in the intima, or single lesion less than one cm in length in the media; 2, multiple punctate lesions of calcification in the intima, or ≤3 lesions with irregular pattern & linear pattern (<6 cm) in the intima, or single lesion less than three cm in length in the media, or combination of these above VCs in the intima and media; 3, >3 lesions with irregular pattern & linear pattern (<6 cm) in the intima, or single calcification lesion ≥6 cm in the intima, or single lesion more than three cm in length or multiple lesions in the media, or combination of these above VCs in the intima and media (*Liu et al., 2012*). The final sum of points for each vascular segment generated the calcification score of the artery, with the scores ranging from 0 to 9 points for each artery. Two researchers reviewed all stored ultrasound images respectively. In the cases with different readings, the ultrasound examination was re-performed by both researchers, and the decision was made on the basis of consensus (Fig. 2B).

We use the following criteria to differentiate the VC severity of an artery: mild, any segment of the vascular had the score of 1, and the total score of an artery was ≤2; moderate, any segment of the vascular had the score of 2, or the total score of an artery was 3 or 4 ($3 \leq$ total score $\leq 4$); severe, any segment of the vascular had the score of 3, or the total score of an artery was more than 4, then the entire vascular was considered severely diseased.

## Demographic information and clinical characteristics

Demographic information and clinical characteristics were collected from all enrolled patients. Data included age, sex, body mass index, hemodialysis duration, AVF duration, primary renal disease, dialysis clearance, and history of diabetes, hypertension, cardiovascular disease, and cerebral stroke.

## Statistical analysis

Statistical analysis was conducted using SPSS 26.0 (IBM SPSS, Armonk, NY, USA). The data of the clinical variables and demographic characteristics are expressed as counts (percentages) for discrete variables, the mean ± standard deviation (SD) for continuous variables with normal distribution, or a 95% confidence interval (CI) for continuous variables with a skewed distribution. Kappa values were calculated to determine the agreement rate with respect to $p < 0.05$.

**Table 1  Demographic and clinical characteristics of all patients.**

| Variables | | All patients ($n = 40$) |
|---|---|---|
| Age (years) | mean ± SD | 54.05 ± 15.18 |
| Gender | male/female | 25/15 |
| BMI | mean ± SD | 22.96 ± 3.89 |
| AVF location | left/right | 28/12 |
| AVF duration (Months) | Median, 95% CI | 32.00, 34.04-58.36 |
| Diabetes | $n$ (%) | 15 (37.5%) |
| Hypertension | $n$ (%) | 34 (85%) |
| Cardiovascular history | $n$ (%) | 2 (5%) |
| Cerebrovascular event history | $n$ (%) | 8 (20%) |
| Smoke history | $n$ (%) | 15(37.5%) |
| Hemodialysis duration (Months) | Median, 95% CI | 46.50, 42.89-76.26 |
| Primary renal disease | $n$ (%) | |
| Glomerulus nephritis | | 21(52.5%) |
| Diabetic nephropathy | | 12 (30.0%) |
| Polycystic kidney disease | | 4 (10.0%) |
| Hypertensive nephropathy | | 3 (7.5%) |

# RESULTS

## Demographic data and clinical characteristics

Among the 40 patients included in the study, the mean age was 54.05 ± 15.18 years, and 62.5% were males (Table 1). The median hemodialysis duration was 46.50 (42.89, 76.26) months. The primary renal disease of the involved patients included diabetic nephropathy ($n = 12$, 30.0%), chronic glomerulonephritis ($n = 21$, 52.5%), hypertensive renal disease ($n = 3$, 7.5%), and polycystic renal disease ($n = 4$, 10.0%). Fifteen patients had diabetes, and 34 patients had hypertension. In addition, two patients had a history of cardiovascular disease, while eight patients had a history of stroke. According to the ultrasound examination results, VC was observed in 31 patients, and the VC on cross sections was consistent with the findings on longitudinal sections. VC was observed in 22 patients, according to the X-ray examination results.

## Utility of X-ray to detect VC lesions

In total, 160 upper extremity arteries (80 radial and 80 ulnar arteries) of 40 patients received ultrasound and X-ray examinations to screen VC pathological lesions. The ultrasound examination detected calcification in 76 (47.5%) of the 160 upper extremity arteries, while the X-ray examination detected calcification in 56 (35%) of the 160 upper extremity arteries. Ultrasound had a higher detection rate, and the proportion of VC arteries detected by ultrasound was higher than that detected by X-ray (Table 2). Ultrasound was more accurate in detecting the presence of calcification than X-ray. According to ultrasound, the sensitivity, specificity, positive predictive value, and negative predictive value of X-ray for the detection of VC were calculated as 73.21%, 66.35%, 53.95%, and 82.14%, respectively. When including all patients, the measurements
**Table 2  Utility of X-ray for the detection of VC in patients.**

| | | X-ray | | |
| --- | --- | --- | --- | --- |
| | | Any VC (number of arteries) | Non VC (number of arteries) | Total (number of arteries) |
| Ultrasound | Any VC (number of arteries) | 41 | 35 | 76 |
| | Non VC (number of arteries) | 15 | 69 | 84 |
| | Total (number of arteries) | 56 | 104 | 160 |

Notes.
   Sensitivity: 73.21%; Specificity: 66.35%; Positive predictive value: 53.95%; Negative predictive value: 82.14%.

**Table 3  Distribution of VC in different segments according to the ultrasound assessment compared to the X-ray findings.**

| VC lesions | Proxima segments (number of arteries) | Middle segments (number of arteries) | Distal segments (number of arteries) | P value |
| --- | --- | --- | --- | --- |
| Ultrasound | 12 | 38 | 66 | 0.235 |
| X-ray | 9 | 44 | 47 | |

**Table 4  Distribution of VC severity according to the ultrasound assessment compared to the X-ray findings.**

| | Mild (number of arteries) | Moderate (number of arteries) | Severe (number of arteries) | P value |
| --- | --- | --- | --- | --- |
| ultrasound | 65 | 11 | 0 | 0.000 |
| X-ray | 25 | 27 | 4 | |

of agreement between ultrasound and X-ray were assessed by Kappa statistics with $k = 0.365$ ($p = 0.000$).

   Furthermore, there was no significant difference in the detection of VC at different segments of arteries between ultrasound and X-ray (Table 3). However, we observed a significant difference in the detection of the severity of VC between ultrasound and X-ray (Table 4), in which ultrasound detected more mild VC lesions compared to X-ray. In addition, more VC lesions were observed in the distal part of arteries(radial and ulnar) compared to that in the middle or proximal part of arteries (Table 3).

   In all the patients, 45 (56.25%) ulnar arteries had VCs, while 30 (37.50%) radial arteries had VCs according to ultrasound assessment. The ulnar arteries had a significantly higher rate of VC than the corresponding radial arteries ($p = 0.013$). Consistently, according to the X-ray assessment, there were 38 (47.5%) cases of ulnar arteries with the presence of VC, which was significantly higher than the 18(22.5%) cases of corresponding radial arteries ($p = 0.001$).

## DISCUSSION

In the present study, we investigated the level of agreement on VC diagnosis between ultrasound and X-ray. Our data have shown that both ultrasound and X-ray are reliable methods to detect the presence of calcification. According to the ultrasound results, X-rays had a sensitivity of 73.21% and a negative predictive value of 82.14%.

Previous studies discovered VC in hemodialysis patients. *Niu et al. (2019)* investigated VC in the medium and small arteries in dialysis patients, such as the femoral artery, radial artery, and digital artery, and found that the calcification rate was 42.9%. *Kraus et al. (2015)* suggested that 77.84% of ESCKD patients had abdominal aortic calcification. Consistently, we observed VC in 77.5% of our patients' radial or ulnar arteries according to the ultrasound findings. In fact, most cases of calcification of medium and small arteries are not solitary lesions, and VC in upper extremity arteries may indicate the existence of VC in other similar arteries, such as the coronary artery. VC in the coronary artery could be associated with myocardial ischemia. *Achim et al. (2022)* reported that VC in radial arteries was associated with calcific coronary plaques. Thus, the progression of VC in upper extremity arteries should be monitored in patients on hemodialysis, and this monitoring may provide more information about calcific coronary plaques.

Although both ultrasound and X-ray are widely used to detect VC in the clinic, the accuracy and consistency of the examination results by these two methods remain unclear, especially in the detection of VC in upper extremity arteries. Our data have shown that ultrasound had greater sensitivity in the detection of VC than X-ray. Ultrasound detected more VC lesions than X-ray in total, especially in the detection of mild VC lesions. Hence, ultrasound might be superior to X-ray in the identification of VC lesions when screening VC in patients. Moreover, ultrasound can distinguish intimal calcification and medial arterial calcification and is more sensitive than plain X-rays (*Jashari et al., 2015*). In contrast, the anatomical structures nearby or overlying the radial or ulnar arteries might affect the ability of X-ray to detect VC in vascular, such as soft tissues. Therefore, false-negative findings might be one of the significant limitations of using X-ray to detect VC. X-rays, as an imaging method to detect VC, have been shown to be less sensitive than ultrasound. *Madden et al. (2007)* investigated extracranial moderate to severe carotid artery calcification using panoramic radiographs, and they reported a high number of false-positives, resulting in low sensitivity and positive predictive values, which indicated that the panoramic radiograph could be an unreliable method to detect calcification. However, some other studies have argued that radiography and ultrasound can reach high agreement in the detection of VC. *Khosropanah et al. (2009)* found that radiographic radiography examinations had a high negative predictive value to detect VC on both sides of the carotid arteries according to the ultrasound results, with $k = 0.61$, $p = 0.001$. *Friedlander et al. (2005)* reported that carotid artery calcification was identified in 65 patients by panoramic radiographs, and all were confirmed to have atherosclerotic lesions by ultrasound scans. Therefore, more research is needed to compare the reliability between ultrasound and X-ray. And a gold standard such as CT or histology examination should be involved to help the researcher to determine the diagnostic accuracy of ultrasound or X-ray.

In addition, our data showed that ulnar arteries had more VC lesions than the corresponding radial arteries in hemodialysis patients. However, the underlying pathological mechanism for unequal VC distribution in different upper extremity arteries remains unknown. We speculated that blood flow, blood pressure, shear stress, and response to calcification inhibitors might play a complex role in VC pathology (*Rocha-Singh, Zeller & Jaff, 2014*; *Schlieper et al., 2016*). Our data indicate that ulnar artery VC might require more attention to explore its clinical implication in hemodialysis patients.

Both US and X-ray data identified more VC lesions in the distal part of the radial artery than in the middle or proximal part of the radial arteries, and similar trends were also observed in the ulnar artery. These results were not consistent with previous studies in the general population. For example, *Bishop et al. (2008)* found more VC lesions in the proximal part of arteries than in the distal part of arteries, but they also found that VC in distal arteries was more related to peripheral arterial disease. In our cohort, all the involved subjects were patients with ESCKD who were receiving hemodialysis, some of whom were complicated with diabetes or hypertension. ESCKD, diabetes, and hypertension are all risk factors that contribute to the calcification of arteries, and VC pathology in arteries might have some difference between hemodialysis patients and the general population. Some preliminary studies have explored the pathology of VC in hemodialysis patients. *Ohtake et al. (2023)* suggested that uremia could deteriorate VC pathology in distal muscular arteries *via* the osteogenic differentiation of vascular smooth muscle cells. VC in the medial layer (known as Monckeberg medial sclerosis) was more frequently observed in hemodialysis patients, while *Blacher et al. (2001)* suggested that VC in the medial layer caused the loss of arterial elasticity, which then increased pulse pressure to the distal arterial, contributing to VC pathology. In fact, multiple studies have reported more calcifications in the distal part of arteries than in other parts of arteries in hemodialysis patients. *Sigrist & McIntyre (2008)* suggested a relatively high incidence of diffuse distal lesions of the lower limb arteries in hemodialysis patients. *Ohtake et al. (2023)* observed more VC lesions in the distal arterial portion of the lower limbs in hemodialysis patients.

Although *Jashari et al. (2015)* suggested that ultrasound is an accurate method to detect calcification in carotid arteries, especially calcification volumes $\geq 8$ mm$^3$, we must admit that we failed to detect arteries with severe calcification by ultrasound. In contrast, four arteries with severe calcification were identified by X-ray in 40 involved patients. The potential reason could be attributed to failure to detect VC in the media by ultrasound. In some cases, likely due to reflected signal processing and spurious linear patterns of echogenicity, medial VC may be masqueraded. The evaluation of calcification may be hampered by acoustic shadowing (*Mohebali et al., 2015*; *Seyman et al., 2019*). *Seyman et al. (2019)* suggested that acoustic shadowing could impair accuracy in the detection of stenosis in carotid arteries by ultrasound, while *Mohebali et al. (2015)* suggested that acoustic shadowing could impair the accurate characterization of calcification plaque in carotid ultrasound examinations. Other reasons could be the poor correlation between ultrasound and vascular histology, inadequate plaque visualization, operator dependence, inconsistent parameters of calcification lesions, and the limitation to a two-dimensional assessment (*Denzel et al., 2004*). In addition, the overestimation on X-ray could not be ruled

out. *Khosropanah et al. (2009)* suggested that carotid calcification could be overestimated by panoramic radiography.

In the present study, approximately 40–70 mins were required to complete the ultrasound examination in one patient. The cross-sectional ultrasound view was used to identify veins and arteries and their orientation to each other. The longitudinal ultrasound view is technically more difficult to obtain. To obtain qualified figures to identify the VCs in the longitudinal plane, the researcher needs to adjust the depth of the ultrasound machine, and light pressure should be applied to the soft tissues because vigorous pressing of the transducer leads to compression of the artery. In addition, we suggested that the examiner partially rest the hands between examinations to maintain consistency. To identify the target artery, pulsed wave Doppler and color flow Doppler are additional tools that can be used in a longitudinal plane at an angle of 45−60° and parallel to the direction of the blood flow/vessel walls. In our study, all ultrasound examinations were performed by a single highly experienced operator following a standard protocol on all patients. This may have minimized confounding bias. In the future, research direction should involve to optimize the ultrasound examination protocol to shorten the examination duration through multiple ways, such as standardize protocols, focus on relevant findings related to vascular calcification, use high-frequency linear probes, select appropriate transducer frequencies, and optimize gain settings.

This was a single-center study, so the generalizability of the results may be limited. Only patients on hemodialysis were involved in this study, which affected the reliability of our findings. Therefore, a control population is needed in future studies. Second, the risk factors for VC in patients were not investigated. Third, our data have shown that ultrasound was more sensitive in detecting mild VC lesions, while X-ray had advantages in detecting moderate and severe VC in upper extremity arteries. However, the gold standard of VC examination, such as CT or histology examination, was not considered in this study, which might jeopardize the results of this study. In addition, the relationship between VC and AVF failure, such as AVF stenosis, nonmaturation, or thrombosis, was not investigated.

## CONCLUSIONS

Ultrasound is a sensitive and reliable method to screen VC in upper extremity arteries. X-ray is not sufficiently accurate to detect early calcification pathology, whereas ultrasound could enable the detection of earlier stages of calcification. The proportion of ulnar artery calcification was higher than that of radial artery calcification, which might need more future investigation. Moreover, the consistency study between ultrasound and X-ray in detecting VC has the potential to guide the selection of the appropriate imaging method for screening artery calcification in hemodialysis patients. It can also assist in determining whether an additional imaging method is needed when using ultrasound or X-ray alone to identify vascular calcification.

### Funding

This project was supported by the Key Research Program of Hebei Province, China (Grant No. 21377747D) and by the Medical Science and Research Program of the Health Commission of Hebei Province, China (Grant No. 20221370). The funders had no role in study design, data collection and analysis, decision to publish, or preparation of the manuscript.

### Grant Disclosures

The following grant information was disclosed by the authors:
Key Research Program of Hebei Province, China: 21377747D.
Medical Science and Research Program of the Health Commission of Hebei Province, China: 20221370.

### Competing Interests

The authors declare there are no competing interests.

### Author Contributions

- Yanli Yang conceived and designed the experiments, performed the experiments, analyzed the data, prepared figures and/or tables, authored or reviewed drafts of the article, and approved the final draft.
- Na Lin conceived and designed the experiments, performed the experiments, analyzed the data, prepared figures and/or tables, authored or reviewed drafts of the article, and approved the final draft.
- Yuankai Xu performed the experiments, prepared figures and/or tables, and approved the final draft.
- Zheli Niu performed the experiments, analyzed the data, prepared figures and/or tables, and approved the final draft.
- Fulei Meng analyzed the data, prepared figures and/or tables, and approved the final draft.
- Kaidi Zhang analyzed the data, authored or reviewed drafts of the article, and approved the final draft.
- Yuhuan Wang analyzed the data, authored or reviewed drafts of the article, and approved the final draft.
- Lin Ruan conceived and designed the experiments, prepared figures and/or tables, authored or reviewed drafts of the article, and approved the final draft.
- Lihong Zhang conceived and designed the experiments, prepared figures and/or tables, authored or reviewed drafts of the article, and approved the final draft.

### Human Ethics

The following information was supplied relating to ethical approvals (*i.e.*, approving body and any reference numbers):

The First Hospital of Hebei Medical University institutional ethics committee approved the study (NO. 20210372).

## Clinical Trial Ethics

The following information was supplied relating to ethical approvals (i.e., approving body and any reference numbers):

The First Hospital of Hebei Medical University institutional ethics committee.

## Data Availability

Raw data, including all patients' demographic and clinical characteristics, are available in the Supplemental Files.

## Clinical Trial Registration

The following information was supplied regarding Clinical Trial registration:

ChiCTR1900021975

## Supplemental Information

Supplemental information for this article can be found online at http://dx.doi.org/10.7717/peerj.15855#supplemental-information.

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
