# Peer review of "Calcification detection on upper extremity arteries: a comparison of ultrasonic and X-ray methods"

_PeerJ, doi:10.7717/peerj.15855_

## Round 0.1 · original submission · Major Revisions

Please clarify the used methodology.

·

Basic reporting

The research topic is significant for the discipline. The research task and problems are clear and carefully justified.

The source material is relevant to the topic and fresh, and the majority of the sources are scientific publications.

The structure is clear and logical, and the research process is easy to follow. The text is written with a fluent academic style and register and is easy to read.

The results have been presented in an organized manner, faultlessly and illustratively. However, the figure 2 legend should be checked as A for x-ray images and B for ultrasound images, and the word (serve) should be changed to severe. Table 1 should be summarized; laboratory values are not needed as all patients categorized as (ESCKD)

The English language should be improved. An example where the language could be improved is the abstract as whole.

Experimental design

Suitable basic methods have been chosen for the research problems, and
they have been used duly.

A sufficient amount of research material has been used in relation to the research task.

The research process has been implemented faultlessly.

Validity of the findings

The results are presented in relation to earlier literature and theoretical viewpoints, but argumentation is mainly declaratory.

All underlying data have been provided; they are robust, statistically sound and controlled.

Conclusions are well stated, linked to original research question.

Additional comments

The discussion and conclusions are anchored to the main results. Discussion may still contain some incomplete and unorganized parts.

The limitation of the ultrasound in detecting severe calcification should be mentioned in the main findings and should be justified in the discussion.

·

Basic reporting

Clear but ambiguous. The methodology requires significant reworking.
Literature references are sufficient.
Professional structure is adequate.
The data requires more careful interpretation, largely based on clarification of methodology.

Experimental design

Original research confirmed.
Relevant questions addressed.
Ethical standards fulfilled, Rigor of interpretation is partly missing, but could be improved by greater attention and clarification of methodology (see below):

Intimal and medial calcifications represent distinct histologic (and pathogenetic) entities each with specific patterns on US and XR as follows: US – intimal: inhomogeneous, spotty, irregular, adluminal, frequent shadows; US – medial: homogeneous, sheets-like, linear, abluminal , rarely shadows; XR – intimal: spotty (if plaque constituent), linear (if diffusely calcified myo-intima); medial: linear, regular pattern. Combinations of patterns are common. The criteria for the definition of VC by each modality is critical to all of the analysis and requires a more detailed description. Thus, the echogenicity of the myointima on US may vary between the near and far walls (less frequently seen in medial VC in the former), and cross-sectional (CS) views may assist in decisions; medial VC rarely displays “shadowing” compared with intimal VC; XR does not allow distinction between intimal and medial VC; given the presence of the threshold of VC XR define reliably the VC, in contrast, VC below the threshold will be missed. Consequently, both techniques using different principles will likely have different strengths and weaknesses in defining VC, depending on the reflexivity (US) and absorption coefficient (XR) of examined tissues. The presented data seems to support the notion that in the majority of patients mild VC was present escaping the detection by XR. In contrast, above the threshold VC, the detection seemed better, when compared to US. Rather surprising appears the ability of US to detect mild VC while missing severe VC. It appears that while intimal VC are typically detected, medial VC, particularly, if mild, can be easily missed. In some cases, likely due to reflected signals processing, spurious linear patterns of echogenicity may masquerade for medial VC. Based on these ambiguities the methodology needs to be described in greater detail addressing the employed definitions and criteria directly affecting results and interpretations.

Validity of the findings

The validity of data in this study depends on tissue typing specific to both modalities (US - ultrasound) and XR (plain X-ray radiography). The methodology defines the validity of the results and needs to be reworked.
The raw data has not been provided. Examples of the scoring systems ( Figure B 2) requires more detailed explanation. It is not clear whether data acquired from cross-sectional images were included in the analysis. No missing or suboptimal quality data was reported. Experience shows that particularly in the upper third of the upper extremity it may be rather difficult to identify both arteries (ulnar and radial).

Additional comments

The limitations and differences between US and XR should be stated (US – mild VC, XR moderate and severe VC, “better” respectively).
Data on cross-sectional US images have not been shown or discussed; was there agreement between those and longitudinal sections?
The difference in VC prevalence between the upper and lower thirds of arteries seems counter-intuitive and should be addressed.
Similarly, the difference between the ulnar and radial arteries in terms of VC prevalence seems surprising. Were both arteries identified by US in all patients equally well in all segments? Frequently, in the upper part of the upper extremities the arteries are more difficult to follow. Could part of the observed differences be due to technical reasons?
The time/patient required to visualize all three parts of the artery requires comment as to the clinical practicability of the authors' suggestion and clinical application.
What is the meaning of p-value in table 3?

Typographical errors and stylistic flaws need to be addressed – selected examples
Some of the authors’ names are capitalized other not.
201 popular pathological change (replace by representing frequent or common pathological findings)
213/214 we have do admit replace e.g. we failed to identify
215 involved, replace by e.g. affected
218 reduction, replace by e.g. limitation
220 Our data have shown that ultrasound had more sensitivity in the detection of VC, compared to more sensitivity replace by e.g. greater
Table 3 Proxima segment

---

## Round 0.2 · Major Revisions

Please address the issue raised by second reviewer carefully.

·

Basic reporting

The authors made substantial changes as requested.

Experimental design

The research question is well defined, relevant, and meaningful. It is stated that research fills an identified knowledge gap.

Validity of the findings

Conclusions are well stated, linked to the original research question, and limited to supporting results.

·

Basic reporting

Adequate in all aspects.

Experimental design

Sound.

Validity of the findings

Detection of VC in the orearm arteries is of clinical interest mainly in patients undergoing encovascular transradial or transulnar procedures. There seems to be littlle or no additional prognostic value benefit cocerning clinical outcomes. Considering the long examination time as reported by the authors, the practicability of the clinical use tends toward zero. The auhtors should address these issues in Discussion still: that is: what is the scientific and clinical benefit. of assessing VC in foream arteries. How the examination time could be shortened if targeting clinical application.

The comments on missing severe VC by ultrasound remain puzzling, particularly given the thorughness of the examination. The authors shoud provide images of those missed severe VC as visualized by X-ray and US, provide criteria applied to assess severe VC on US - images and instrumentation's settings.

Additional comments

Second revision addressing the open issues is requrested.

---

## Round 0.3 · accepted · Accept

The reviewers' concerns were adequately addressed.